# Prokaryotic Expression and Functional Verification of Antimicrobial Peptide LR_GG_

**DOI:** 10.3390/ijms25137072

**Published:** 2024-06-27

**Authors:** Xiang Liu, Yining Ding, Yuhan Shen, Sizhuo Liu, Yuehua Liu, Yuting Wang, Shikun Wang, Claudio Orlando Gualerzi, Attilio Fabbretti, Lili Guan, Lingcong Kong, Haipeng Zhang, Hongxia Ma, Chengguang He

**Affiliations:** 1Engineering Research Center, The Chinese Ministry of Education for Bioreactor and Pharmaceutical Development, College of Life Sciences, Jilin Agricultural University, Changchun 130118, China; liuxiang_0915@163.com (X.L.); wangyuting990501@163.com (Y.W.);; 2Westlake Laboratory of Life Sciences and Biomedicine, Hangzhou 310024, China; 3School of Biosciences and Veterinary Medicine, University of Camerino, 62032 Camerino, Italy

**Keywords:** antimicrobial peptides, prokaryotic expression, antibacterial mechanism

## Abstract

The antimicrobial peptide LR_GG_ (LLRLLRRGGRRLLRLL-NH2) was designed and chemically synthesized in a study conducted by Jia et al. Gram-negative bacteria were found to be sensitive to LR_GG_ and exhibited a high therapeutic index. Genetic engineering methods were used to create the prokaryotic fusion expression vector pQE-GFP-LR_GG_, and the resulting corresponding fusion protein GFP-LR_GG_ was subsequently expressed and purified. The precursor GFP was then removed by TEV proteolysis, and pure LR_GG_ was obtained after another round of purification and endotoxin removal. The prokaryotic-expressed antimicrobial peptide LR_GG_ displays a broad-spectrum antibacterial effect on Gram-negative bacteria, and its minimum inhibitory activity (MIC) against Escherichia coli can reach 2 μg/mL. Compared to the chemically synthesized LR_GG,_ the prokaryotic-expressed LR_GG_ exhibits similar temperature, pH, salt ion, serum stability, and cell selectivity. Furthermore, prokaryotic-expressed LR_GG_ showed excellent therapeutic effects in both the infection model of cell selectivity and no embryotoxicity in a *Galleria mellonella* infection model. The mechanism by which LR_GG_ causes bacterial death was found to be the disruption of the Gram-negative cell membrane.

## 1. Introduction

Untreatable multidrug-resistant infections are becoming more common, causing an alarming number of deaths [1]. To address this issue, scientists are exploring alternative methods to develop drugs that can target cell components or mechanisms that are less prone to mutation, resulting in resistance [2,3]. Among the new antimicrobial substances being developed, antimicrobial peptides (AMPs) show great potential as a new source of effective antimicrobial activity [4,5,6].

AMPs, also known as host defense peptides, are the oldest innate immune defense factors [7] and play a crucial role in the natural immune system of the host [8]. Recent studies have revealed that AMPs use unique antimicrobial mechanisms that differ from those of the antibiotics commonly used in clinical infections [9,10,11]. There are a variety of AMPs that bind to microbes through low-affinity targets, a condition that greatly reduces the chance of generating antimicrobial resistance [12]. AMPs are categorized according to their effect on the bacterial membrane or membrane proteins [13,14,15,16,17]. Under stress conditions, the occurrence of mutations in these areas is reduced, thus decreasing the chance of developing resistance [17,18,19,20,21,22]. These AMPs are the original innate immune defense factors, also known as host defense peptides [21]. They are crucial to the host’s innate immune response [22]. Research has shown that AMPs have a unique mechanism of action against microorganisms, unlike the current antibiotics used to treat infections. They have multiple low-affinity targets with microorganisms, and because of its fast degradation rate, it will not produce drug residues, which reduces antibiotic resistance. In light of these properties, AMPs have the potential to be used as drugs to treat infections caused by microorganisms [23,24,25].

It has been shown that the designed AMP LR_GG_ (LLRLLRRGGRRLLRLL-NH2), which tends to form an α-helical structure in membrane-mimetic environments [26], exhibits effective antimicrobial activity against Gram-negative bacteria when used in combination with ciprofloxacin, colistin, and other antibiotics. This combination has been found to reverse the ciprofloxacin-resistant phenotype of *Pasteurella multocida* and increase 512-fold the antibacterial activity of ciprofloxacin against this bacterium (unpublished data from our laboratory).

The chemically synthesized peptide LR_GG_ has been found to damage the bacterial outer membrane by targeting lipopolysaccharide (LPS) and disrupting the Fe^3+^ transport system involving the inner membrane permease protein FecD, thereby significantly improving the efficacy of ciprofloxacin in the treatment of animals infected with multidrug-resistant *Escherichia coli* and *Pasteurella multocida*. The LR_GG_ peptide can also trigger membrane potential hyperpolarization and intracellular accumulation of reactive oxygen species (ROS) in pathogenic bacteria, leading to synergistic bactericidal effects. However, neither the isolation nor the chemical synthesis of AMPs is a good choice for clinical application. In fact, the isolation of natural AMPs from hosts is impractical due to their low concentration and complicated purification procedure, while the use of synthetic AMPs is limited by the high costs of their chemical synthesis.

Currently, the most effective strategy to produce AMPs on a large scale is biological expression [27,28], with the expression in *Escherichia coli* being the most commonly used method in light of its low production costs, short incubation period, and ease of control [29,30]. However, the production of AMPs composed of only ten to a dozen amino acids in prokaryotic expression systems can be challenging [31]. An additional drawback is that AMPs produced within prokaryotes can result in cell inhibition. To avoid this problem and increase the AMPs production rate, a larger protein tag, such as green fluorescence protein (GFP), can be fused to AMPs [32,33]. GFP is commonly used as a protein label and can also be used for easily observable expression detection [34,35].

In this study, the GFP was tandemly fused with the N′ terminus of AMP LR_GG_ and cloned and inserted into the pQE80 *Escherichia coli* expression vector to act as a precursor protein, and a target site for the TEV protease was inserted between the GFP fusion protein-encoding gene and the LR_GG_, thus enabling the fusion label to be easily removed during purification. pQE80 is a vector plasmid used for protein expression, hosted by *Escherichia coli*, having the prokaryotic resistance of Kanamysin and being a promoter of T5, suitable for expressing short peptides. The cleavage site affects only one amino acid of the main protein chain with minimal impact on the protein properties [36]. In this study, the effects of AMP LR_GG_ expression on the cell membrane and DNA of Gram-negative bacteria were also investigated through tests of the permeability of the inner and outer membranes, cell membrane potential, and electrophoretic mobility shift assay (EMSA).

## 2. Results

### 2.1. Construction of the pQE-GFP-LR_GG_ Expression Vector

Agarose gel electrophoresis analysis of the linearized pQE80-KAN vector, GFP gene, and TEV site LR_GG_ gene sequences confirmed that their fragment lengths were 5329 bp, 719 bp, and 72 bp, respectively (Figure 1A). PCR identification of the transformed competent clones using the primers pQE30+ and pQE30− revealed the expected band size of 961 bp for the GFP + LR_GG_ + TEV restriction sites, as shown in Figure 1B. These results validated the successful construction of the prokaryotic expression vector pQE-GFP-LR_GG_.

### 2.2. Expression and Purification of the Fusion Protein GFP-LR_GG_

The expression and purification of all the samples were verified through SDS–PAGE, and the corresponding results are shown in Figure 2. Notably, the target band measuring 31 kDa was found in lane 3, indicating that GFP-LR_GG_, the intended protein, could be eluted with 300 mM imidazole. Additionally, the concentration of the purified target protein was quantified to be 4.7 mg/L using a BCA kit.

### 2.3. Cutting of the Fusion Protein and Purification of the Antibacterial Peptide LR_GG_

The recombinant TEV enzyme cleaved the GFP, which was subsequently purified using a Ni-NTA column and analyzed via Tricine-SDS–PAGE electrophoresis. In Figure 3, the results show that the strip in lane 2 is slightly lower than the pre-cut strip in lane 1 and has the same height as the strip passing through the liquid. This indicates that the fusion label was successfully removed, leaving behind a band with a minimal molecular weight consistent with the chemical synthesis of LR_GG_ in lane 4. To summarize, the recombinant TEV enzyme cleaved the GFP, and the prokaryotic-expressed antimicrobial peptide LR_GG_ was successfully purified through multiple purification methods. The concentration of the protein, as measured by the BCA reagent kit, was an average of 1.4 mg/L.

### 2.4. Determination of the Bacteriostatic Activity and Kinetics Curve of Fusion-Expressed Antimicrobial Peptide LR_GG_

This study employed the microdilution method to assess the minimum inhibitory concentrations of both the expressed and synthetic LR_GG_ peptides. Table 1 presents the results, which reveal that LR_GG_ antimicrobial peptides demonstrate exceptional broad-spectrum antibacterial activity against Gram-negative bacteria, surpassing their efficacy against Gram-positive bacteria. Notably, the MIC of prokaryotic-expressed and chemically synthesized antimicrobial peptides were found to be similar, while the fusion protein showed no antibacterial activity.

### 2.5. Environmental Sensitivity of Fusion Expressed Peptide LR_GG_

Temperature and pH stability: Based on the data presented in Table 2, both the expressed and chemically synthesized LR_GG_ demonstrated remarkable temperature and pH stability. Specifically, the MIC value of these peptides increased by four-fold at 100 °C and by two-fold and four-fold at pH 10. These findings demonstrate that LR_GG_ exhibits excellent stability under conditions that are typically considered harsh, highlighting its potential as a promising candidate for various applications.

According to the data presented in Table 3, the MIC values of melittin was substantially greater than the MIC values of protease-treated melittin. This finding highlights the importance of proteases in reducing the stability of melittin. Furthermore, it has been reported that the peptide LR_GG_ is unstable in the presence of proteases.

Salt Ion Stability: Given the abundance of salt ions present in physiological environments, it is important to assess the MIC values of the antimicrobial peptide LR_GG_ when expressed versus when chemically synthesized in various salt ion environments. As demonstrated in Table 4, compared with those in the control, the MICs of the prokaryotic-expressed and chemically synthesized peptides in a 150 mM NaCl solution decreased by only two- to four-fold. Therefore, it can be concluded that both the expressed and chemically synthesized peptide LR_GG_ possess commendable salt ion stability.

Serum stability: The MIC for peptide LR_GG_, both expressed and chemically synthesized, were assessed against *Eschericha coli* ATCC25922 at various serum concentrations. The results are presented in Table 5. Notably, the MIC for LR_GG_ fluctuates at serum concentrations of 20% to 50%, ultimately increasing four-fold. Conversely, the MIC values for melittin plateaued at 128 at a 10% concentration, demonstrating minimal antibacterial activity. These findings suggest that the serum environment has little impact on the antibacterial effectiveness of LR_GG_, regardless of its method of synthesis.

### 2.6. Cytotoxicity and Hemolytic Activity of Antimicrobial Peptide LR_GG_

The effectiveness of the expressed and synthetic peptides LR_GG_ against RAW264.7 and Vero cell was evaluated, as demonstrated in Figure 4). At MIC values of 1–64, the chemically synthesized antimicrobial peptide exhibited a slightly higher cell survival rate compared to the prokaryotic-expressed antimicrobial peptide. However, both the expressed and synthetic peptides showed over 80% cell survival rate against RAW264.7 and Vero cells, in contrast to the control melittin. Compared with that of the melittin control, the hemolytic activity of chemically synthesized LR_GG_ was less than 10% compared to the melittin control at MIC values of 1–64 μg/mL, as shown in Figure 5. The hemolytic activity of the expressed LR_GG_ was slightly more than 10% at 64 μg/mL, but the hemolytic index was less than 10% at the lowest inhibitory concentration. When the MIC value is 128 μg/mL, the hemolytic activity of the expressed LR_GG_ is slightly higher than 20%, which may have slight toxicity [37]. The comparable cytotoxicity and hemolytic activity of LR_GG_-expressed and chemically synthesized LR_GG_ demonstrated that the antimicrobial peptide LR_GG_ has excellent cell selectivity at minimum inhibitory concentrations.

### 2.7. Embryotoxicity of the Antimicrobial Peptide LR_GG_ in Zebrafish

The results of embryotoxicity tests conducted on zebrafish that were exposed to LR_GG_ expression and chemically synthesized antimicrobial peptide are shown in Figure 6. The hearts of zebrafish that were exposed to LR_GG_ expression (a positive control) with high concentrations of sodium dehydroacetate (200 μg/mL) showed significant hemorrhage and severe bending of the fish. Conversely, exposure to expressed LR_GG_ and chemically synthesized LR_GG_ at a concentration of 1 × MIC, comparable to that of the negative control, did not change the morphology of the zebrafish. These results indicate that neither the expressed nor the chemically synthesized antimicrobial peptide LR_GG_ had any toxic effects on zebrafish.

### 2.8. Inner and Outer Membrane Permeability Tests

As depicted in Figure 7A, the permeability of the inner membrane of the recombinant antimicrobial peptide LR_GG_ increased within 0–50 min, and further increased with increasing dose and prolonged exposure. These results indicate that the prokaryotic-expressed antimicrobial peptide LR_GG_ effectively ruptures the inner membrane of bacterial cells at the minimum inhibitory concentration. Additionally, as illustrated in Figure 7B, the peptide was able to penetrate the outer membrane of cells in a concentration-dependent manner at concentrations ranging from 1 to 32 μM. Notably, when the concentration of LR_GG_ exceeded 8 µM, the outer membrane permeability was observed to be more than 80%. These findings suggest that the prokaryotic-expressed antimicrobial peptide LR_GG_ can effectively destroy the bacterial cell outer membrane at the minimum inhibitory concentration.

### 2.9. Effect of the Antimicrobial Peptide LR_GG_ on the Bacterial Plasma Membrane Potential

According to Figure 8, when the diSC3-5 fluorescent dye enters the cell membrane, it can form non-luminescent polymers [38]. However, if the cell membrane is destroyed, the previously entered diSC3-5 will flow out and be detected through its fluorescence value. Over time and concentration within 0–1500 s at 1–8 MIC, the fluorescence value of prokaryotic-expressed antimicrobial peptide LR_GG_ gradually increased, indicating its impact on the bacterial cytoplasmic membrane potential. This further confirmed that the expressed peptide LR_GG_ is capable of destroying the bacterial cell membrane.

### 2.10. DNA Gel Retardation Assay

The determination of LR_GG_’s binding to DNA can be determined through a DNA gel retardation assay, as depicted in Figure 9A. Through this assay, it was discovered that LR_GG_ effectively binds to the genomic DNA of *Eschericha coli* ATCC *25922* and impedes its migration toward the positive pole when the peptide concentration reaches 256 μM. In addition, as shown in Figure 9B, LR_GG_ effectively blocked plasmid DNA migration toward the positive pole only when the peptide concentration reached 128 μM.

## 3. Discussion

Natural AMPs are present at low concentrations in host organisms and are challenging to isolate due to complicated purification procedures. The high cost of chemical synthesis also limits the use of synthetic AMPs, making neither isolation nor chemical synthesis of AMPs good choices for clinical application. Therefore, this study aimed to produce and test the antimicrobial properties of a fully designed AMP that effectively fights against Gram-negative bacteria, which is the peptide LR_GG_.

The mode of expression of AMPs and protein tag fusion plays a crucial role in bacterial expression systems. This affects their subsequent purification. In a study by Chen Xin et al., four fusion tags (TrxA, SUMO, protein internal peptides, and GST) were fused with Mycelin and expressed in *Escherichia coli* for purification and antimicrobial activity assays [39]. Ali et al. fused GFP with the cell insect toxin Cit1a for expression in silkworms. This finding is consistent with the use of GFP and antimicrobial peptide fusion for expression. This approach effectively prevents the bactericidal effects of antimicrobial peptides on the host bacterium and allows further purification of the antimicrobial peptide [40]. The 6X His tag added in this study can be used to purify antimicrobial peptides. The construction method used in this study combines the fusion tag GFP with the TEV restriction enzyme and adds the antimicrobial peptide LR_GG_, which has been reported in previous research. In this study, the expression vector pQE-GFP-LR_GG_ was constructed, and the GFP-LR_GG_ fusion protein was produced by prokaryotic expression. After chromatographic purification and TEV protease cleavage, the purity-expressed LR_GG_ was prepared.

Furthermore, the effects of the MIC and different environmental conditions on the antimicrobial peptide LR_GG_ were examined. The findings demonstrate that LR_GG_ exhibits excellent stability under conditions that are typically considered harsh, highlighting its potential as a promising candidate for various applications. However, the results showed that the stability of the peptide LR_GG_ toward proteases could be improved. This is particularly important because the body contains multiple proteases that can break down proteins. The peptide segment of LR_GG_ is short, consisting of only 16 amino acids, making it highly susceptible to proteolytic degradation. Consequently, the MIC decreased significantly, while the antibacterial activity decreased after protease treatment. Researchers have suggested that modifying the chemical structure of AMPs can enhance the stability of their polypeptides, preventing them from being degraded by proteins [41]. Additionally, the use of liposome-coated AMPs for delivery can also maintain their stability and reduce their toxicity [42]. This approach has potential clinical benefits for the application of antimicrobial peptides.

In vitro experiments were conducted to measure the cytotoxicity and hemolytic activity of chemically synthesized and expressed LR_GG_. The results of safety assessment experiments at the cellular level revealed that LR_GG_ exhibited no hemolytic activity, and no cytotoxicity was detected for either chemically synthesized or expressed LR_GG_ at MIC values of 1–64. Therefore, further research was conducted to determine the toxicity of LR_GG_ to zebrafish embryos. The results showed that the expression of LR_GG_ and chemical synthesis of LR_GG_ at a concentration of 1 × MIC, comparable to the negative control, did not change the morphology of the zebrafish. These results indicate that neither the expressed nor chemically synthesized antimicrobial peptide LR_GG_ had any toxic effects on zebrafish.

The structure of AMPs is crucial to their biological function. The antimicrobial peptide LR_GG_ examined in this study has an amino acid sequence of (LLRLLRRGGRRLLLLRLL-NH2) and is composed of an α-helix structure [26]. With arginine as a positively charged amino acid and leucine as a hydrophobic amino acid, it has a tremendous membrane-breaking structure. The positive charge of this antimicrobial peptide and the negative control on the bacterial membrane increase membrane permeability due to electrostatic interactions. This leads to the release of the antimicrobial peptide LR_GG_ into the cell plasma membrane, eventually causing lysis of the plasma membrane and leading to the death of microbial pathogens. In addition, bacteria can be inhibited by disrupting bacterial cell membranes. In another study, LR_GG_ was shown to be even more effective when used in combination with ciprofloxacin, colistin, and other antibiotics. LR_GG_ damages the outer membrane of pathogenic bacteria by targeting LPS and disrupting the ability of the Fe^3+^ transport system to permease the FecD protein in the inner membrane (unpublished data). In this study, the antimicrobial peptide LR_GG_ only bonded to DNA at high concentrations, not at the MIC. We suspect that this could be because the DNA double-strand itself is negatively charged, and the antimicrobial peptide is positively charged. As the concentration of antimicrobial peptide increases, it will naturally bind to DNA. Therefore, the exact antibacterial mechanism of the antimicrobial peptide LR_GG_ after rupture at the lowest inhibitory concentration requires further study.

## 4. Materials and Methods

### 4.1. Strains and Plasmids

The chemically synthesized AMP LR_GG_ (LLRLLRRGGRRLLRLL-NH2) is a fully designed peptide, purchased from Sangong Co., Ltd. Shanghai, China.

### 4.2. Acquisition of Target Genes

(1)To target the GFP gene, we designed GFP-F and GFP-R primers (as show in Table 6) from the plasmid pTZ18U-GFP previously constructed in the laboratory. PrimeStar Max DNA Polymerase was used to amplify the GFP gene via PCR, with the plasmid pTZ18U-GFP as (show in Table 7) serving as the template.(2)To target the gene sequence of the antimicrobial peptide LR_GG_, the primers LR_GG_-F1 and LR_GG_-R1 were designed. The seamless cloning method were followed to construct the plasmids, as shown in Figure 10.

### 4.3. Construction of pQE-GFP-LR_GG_ Expression Vector

To construct the pQE-GFP-LR_GG_ vector, three gene fragments were amplified.

(1)The *gfp* gene was amplified by PCR reaction using the primers GFP-F and GFP-R, and the pTZ18U-GFP plasmid served as the template.(2)To synthesize the Tev site and LR_GG_ sequence, two oligonucleotides (TEV-LR_GG_-F2 and TEV-LR_GG_-R2) were mixed at a 1:1 ratio in a PCR reaction buffer. The mixture was denatured at 95 °C and annealed by reducing the temperature by one degree per minute from 95 °C to 25 °C within 70 min. This process yielded a double-stranded LR_GG_ DNA sequence containing the TEV target site.(3)The linearized pQE80 vector was amplified by PCR using the pQE-VT-F and pQE-VT-R primers and the pQE80-KAN plasmid as a template.

The LR_GG_ expression vector was created using the seamless cloning technique. In this process, the *gfp* gene fragment was placed in front of the fusion protein. A TEV cleavage site was added between the *gfp* and the LR_GG_ fragment. These fragments were then placed behind the T5 promoter, and the His tag sequence was located in the pQE80-KAN vector. These genes were fused and connected through approximately 15 bp of complementary gene fragments. To carry out seamless cloning, a 20 μL reaction mixture consisting of 100 ng of *gfp* fragment, 50 ng of Tev site-LR_GG_ fragment, 100 ng of linearized pQE80-KAN vector, and 4 μL of seamless enzyme was prepared. The reaction mixture was incubated for 15 min at 50 °C and then placed on ice. The construct was then transformed into Stellar competent cells (Takara). Stellar competent cells have high transformation efficiency and are especially suitable for the preparation of high copy plasmids. Following the manufacturer’s protocol, the cells were plated onto LB plates containing 50 μg/mL kanamycin and incubated overnight. Positive clones were screened using the PCR reaction of test primers pQE30+ and pQE30− and DNA sequencing. The positive clone was then grown, and the expression vector was extracted and transformed into Transgene (DE3) competent cells.

### 4.4. Expression and Purification of Fusion Protein GFP-LR_GG_

The transformed clone, verified by sequencing, was placed in 20 mL of LB medium containing 50 μg/mL kanamycin, centrifuged at 180 rpm, and incubated overnight at 37 °C. When the optical density reached 0.5, protein expression was induced by 0.2 mM Isopropyl β-D-Thiogalactoside (IPTG). After an additional 4 h of incubation, the cells were collected by centrifugation at 8000 rpm at 4 °C and the cell pellet was stored at −80 °C.

To obtain the GFP-LR_GG_ fusion protein, the *Eschericha coli* cells were lysed using a sonicator at 18 W for six rounds of 30 s bursts in Buffer A (20 mM Tris–HCl pH 7.4, 5% glycerol, 500 mM NaCl). The samples were supplemented immediately before use with 6 mM β-Mercaptoethenol, 0.1 mM Benzamidine, and 0.1 mM Phenylmethanesulfonyl fluoride (PMSF). The lysate was passed through a Ni-NTA 6FF column. Proteins with no affinity for the column were eluted with Buffer A containing 50 mM imidazole, whereas the fusion protein was eluted with Buffer A containing 300 mM imidazole. The fractions containing the fusion protein detected by SDS-PAGE electrophoresis were pooled and dialyzed against buffer A to eliminate imidazole, and the fusion protein was concentrated, analyzed with a Bicinchoninic Acid Assay (BCA) test kit, and stored at −80 °C.

### 4.5. Cleavage of the Fusion Protein and Purification of the Antimicrobial Peptide LR_GG_

The TEV protease was used to cleave the GFP-LR_GG_ fusion protein [43] in a reaction mixture consisting of 100 μL of 10× TEV Buffer supplemented with 50 μL of recombinant TEV protease (1 mg/mL from Beyotime Co., Ltd., Shanghai, China) and 40 μg of the fusion protein GFP-LR_GG_ (1 mg/mL). After 16 h of incubation at 16 °C, the 6× His-tag remained upstream of the GFP, while no His-tag remained linked to LR_GG_. Using a Ni-NTA affinity column, the two products of the digestion of the fusion protein were separated by passage through an affinity column that was able to retain the 6× His-GFP while allowing the LR_GG_ peptide to pass through upon elution with 20 mM Tris-HCl (pH 8.0) buffer containing 200 mL of glycerol, 500 mM sodium chloride, and 300 mM imidazole. The effluent containing GFP-LR_GG_ was dialyzed against buffer A containing 50% glycerol. After checking its purity by Tricine-SDS–PAGE, the protein was stored at −80 °C.

### 4.6. Determination of the Antibacterial Activity of Fusion Expressed Antimicrobial Peptide LR_GG_

The minimal inhibitory concentrations against various bacteria displayed by peptides expressed in prokaryotes and chemically synthesized were determined by the broth microdilution method. Two replicates were set up by placing 100 μL of the two antimicrobial peptides (approximately 1024 μg/mL each) in the first rows of the 96-well plates, followed by serial two-fold dilutions to achieve final peptide concentrations ranging from 512 to 1 μg/mL. The strains kept at −80 °C were reactivated, plated, and incubated overnight at 37 °C. Single colonies were picked, inoculated into 5 mL of MH medium, and cultured until they reached OD_600_ = 0.5. The bacterial suspensions were then diluted to 1 × 10^5^ CFU/mL, and 50 μL was added to wells 1–11. Then, 50 μL and 100 μL of deionized water were added to wells 11 and 12, respectively, to provide a positive and a negative control. The concentration of peptide present in the first non-turbid well in which bacterial growth was inhibited was taken as the minimal inhibitory concentration (MIC). This experiment was repeated three times.

### 4.7. Determination of the Bactericidal Kinetic Curve of Fusion Expressed Antimicrobial Peptide LR_GG_

To determine the bactericidal kinetic curve of in vivo-expressed and synthetic peptide, the bacterial suspension was mixed with two peptides to obtain a final peptide concentration corresponding to 4 × MIC. The plate was then shaken at 37 °C and 180 rpm, taking 50 μL aliquots of the bacterial suspension every 20 min and diluting them two-fold with sterile PBS buffer before they were spread evenly on LB-containing solid media and incubated overnight at 37 °C (12–16 h). The number of colonies on each plate was counted, and the number of colonies at 30–300 was taken as the control group. The group not exposed to the antimicrobial peptide was used as a control. The experiment was repeated three times, and the bactericidal kinetic curve was plotted.

### 4.8. Environmental Sensitivity of Fusion Expressed Antimicrobial Peptide LR_GG_

The efficacy of the chemically synthesized and in vivo-expressed LR_GG_ peptide were tested for their efficacy against *Eschericha coli* ATCC 25922 under various conditions, such as heat and different pH conditions, before determining their MIC in triplicate. In one case, the peptides were placed in ice-cold water at 0 °C and heated to 37 °C and 100 °C for 30 min each, and their efficacy was compared to that of an untreated control group. This process was repeated three times. In another case, the peptides were exposed for one hour to pH values of 4, 6, 8, and 10, and their activity was compared to that of an untreated control group. In another set of experiments, the peptides were mixed and incubated with 1 mg/mL of four different proteases. After inactivation of the enzymes, the peptides were tested for their efficacy against *Eschericha coli* ATCC 25922. The peptides were also tested after mixing with various salt solutions and concentrations of fetal bovine serum at concentrations ranging from 5% to 50%. In all cases, the MIC were determined in triplicate using untreated antimicrobial peptides as controls.

### 4.9. Cytotoxicity Assay of the Antimicrobial Peptide LR_GG_

The cytotoxicity of both in vivo-expressed and synthesized LR_GG_ peptides was determined using the Cell Counting Kit-8 (CCK-8) colorimetric assay on RAW 264.7 and Vero cells. To ensure consistency, before the addition to the cell culture plate the in vivo-expressed peptide was diluted with synthetic peptide LR_GG_ using deionized water to achieve a 1–128 μg/mL concentration gradient. The peptides were then incubated with the cells for 16 h at 37 °C in a 5% CO_2_ incubator, with two sets of replicates for each assay. After incubation, 10 μL of 10% CCK solution was added to each well and incubated for 4 h at 37 °C in a 5% CO_2_ incubator. The OD_450_ was measured, and the process was repeated three times for accuracy.

### 4.10. Hemolytic Activity Assay of the Antimicrobial Peptide LR_GG_

To obtain a suspension containing 2% red blood cells, 15 mL of sheep blood was collected in a tube containing 0.2% sodium heparin anticoagulant and centrifuged at 1000 rpm for 10 min. After discarding the supernatant, the red blood cell pellet was collected, washed twice with sterile phosphate buffered saline (PBS), and resuspended in PBS. The expressed peptide was then diluted with deionized water containing synthetic LR_GG_ to achieve a final concentration with the same gradient as the MIC value (1–128 μg/mL). After the erythrocyte suspension was incubated with the antimicrobial peptide LR_GG_ for one hour at 37 °C, the mixture from each well of the 96-well plate was aspirated and centrifuged at 1000× *g* for 10 min. The absorbance of the supernatant was then measured at 570 nm, with three repeats and averaging. Finally, the hemolytic index was calculated as follows: Hemolytic index (%) = (O.D. peptide − O.D. PBS)/(O.D. TritonX − 100 − O.D. PBS) × 100%.

### 4.11. Embryotoxicity of the Antimicrobial Peptide LR_GG_ in Zebrafish

In this experiment, AB-strain zebrafish embryos were introduced to express and synthesize the antimicrobial peptide LR_GG_ at a concentration of 1 × MIC. A sodium dehydroacetate solution of 200 μg/mL served as the positive control, while the negative control group contained only culture medium. The embryos were cultured in a constant temperature incubator at 28 °C for approximately 2–3 days, and their morphology after fertilization was observed using a light microscope.

### 4.12. Inner and Outer Membrane Permeability Tests

To measure the effectiveness of the antimicrobial peptide LR_GG_, bacterial suspensions were prepared with 1.5 mM O-Nitrophenyl β-D-galactopyranoside (ONPG) buffer at an ABS O.D. of 600 = 0.5. The experiment was conducted by adding varying concentrations of the expressed antimicrobial peptide LR_GG_ (1/2 − 4 MIC) to the same volume of *Eschericha coli* ATCC25922 bacterial solution, successively plated in 96-well plates. PBS was added to the control group. The absorbance was then measured at ABS O.D. = 420 nm using a microplate reader at 5-minute intervals for 50 min.

To determine the permeability of the bacterial membrane, N-Phenyl-1-naphthylamine (NPN), a fluorescent dye that cannot enter a bacterial membrane with an intact structure, was used [44]. First, black 96-well plates were used, and the same volume of bacterial suspension and different concentrations of expressed antimicrobial peptides were added. PBS was added to the control group, and 1 mM NPN fluorescent dye was added to reach a final concentration of 10 μM. The plates were then incubated in the dark at room temperature for 30 min, and their O.D. values were measured with a fluorescence microplate reader using an excitation wavelength of 350 nm and an emission wavelength of 420 nm.

The formula used to calculate the outer membrane permeability was
NPN absorption (%) = (F_obs_ − F_0_)/(F_100_ − F_0_) × 100%,
where F_obs_ is the fluorescence value measured in the presence of antimicrobial peptides and F_0_ is the fluorescence value of the negative control. F_100_ is the fluorescence value of the positive control. The experiment was conducted in triplicate for each group.

### 4.13. Effect of the Antimicrobial Peptide LR_GG_ on the Bacterial Plasma Membrane Potential

To determine whether the expressed antimicrobial peptide LR_GG_ can break bacterial cytoplasmic membranes, diSC_3_-5 was used as a fluorescent dye to monitor changes in membrane potential. A bacterial suspension was prepared and adjusted to an ABS O.D. of 600 = 0.5. Next, the same volume of bacterial suspension and varying concentrations of expressed antimicrobial peptide LR_GG_ (1–8 MIC) were added to black 96-well plates. A control group was established using PBS, and diSC_3_-5 dye was added to a final concentration of 0.4 μM and incubated at room temperature in the dark for one hour. Changes in absorbance values between 0 and 1500 s were measured using a fluorescence microplate reader at excitation (622 nm) and emission (670 nm) wavelengths.

### 4.14. DNA Gel Retardation Assay

The *Eschericha coli* ATCC 25922 genome was extracted, and its concentration was determined to be OD260/OD280 = 1.8~2.0. Genomic DNA (400 ng) and the expressed antimicrobial peptide LR_GG_ at final concentrations ranging from 1~512 μM were incubated at 37 °C for 1 h and verified by 1% agarose gel electrophoresis. The experimental data obtained in this study were statistically analyzed using GraphPad Prism 8.0 software. The data are presented as the mean and standard deviations.

The genome of *Eschericha coli* ATCC 25922 was isolated, and its concentration was determined to be OD260/OD280 = 1.8~2.0. Genomic DNA (400 ng) and the expressed LR_GG_ were incubated at 37 °C for one hour at final concentrations in the range of 1~512 μM. The results were verified via 1% agarose gel electrophoresis. To analyze the data obtained in this study, GraphPad Prism 8.0 software was used for statistical analysis. The data are presented as the mean and standard deviations.

## 5. Conclusions

This study has uncovered valuable insights into the antimicrobial functional peptide LR_GG_, which is expressed through the prokaryotic expression vector pQE-GFP-LR_GG_. This study demonstrated that the fusion protein of AMP LR_GG_ has a wide range of antibacterial effects on Gram-negative bacteria. Additionally, the peptide has shown remarkable stability under various environmental conditions, such as temperature, pH, salt ion, and serum conditions. Moreover, this peptide has been proven in tested animal models to be safe at its minimal inhibition concentration, which is encouraging for prospective antibacterial treatments. Under concentration-dependent conditions, LR_GG_ expressed in prokaryotes can cause damage to the inner and outer membranes of Gram-negative bacteria, as well as affect the cytoplasmic membrane potential, leading to their death. This groundbreaking development in the realm of antibacterial treatments holds great potential.

## Figures and Tables

**Figure 1 ijms-25-07072-f001:**
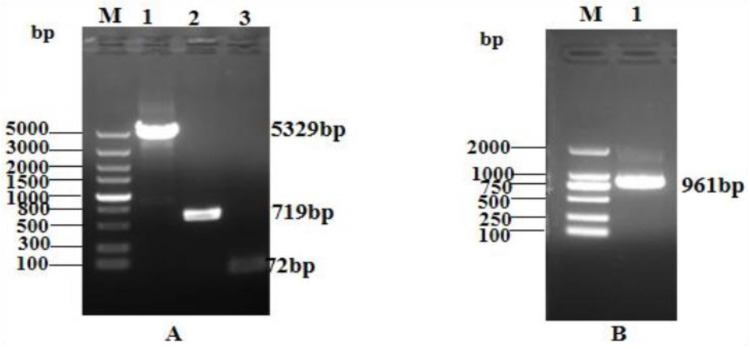
Construction of PQE-GFP-LR_GG_ expression vector. (**A**) lane M: Trans5K DNA Marker; lane 1: Linearized pQE80-KAN vector; lane 2: gfp gene; lane 3: TEV site—LR_GG_ fragmen. (**B**) lane M: Trans2K DNA Marker; lane 1: GFP + TEV sites restriction + LR_GG_.

**Figure 2 ijms-25-07072-f002:**
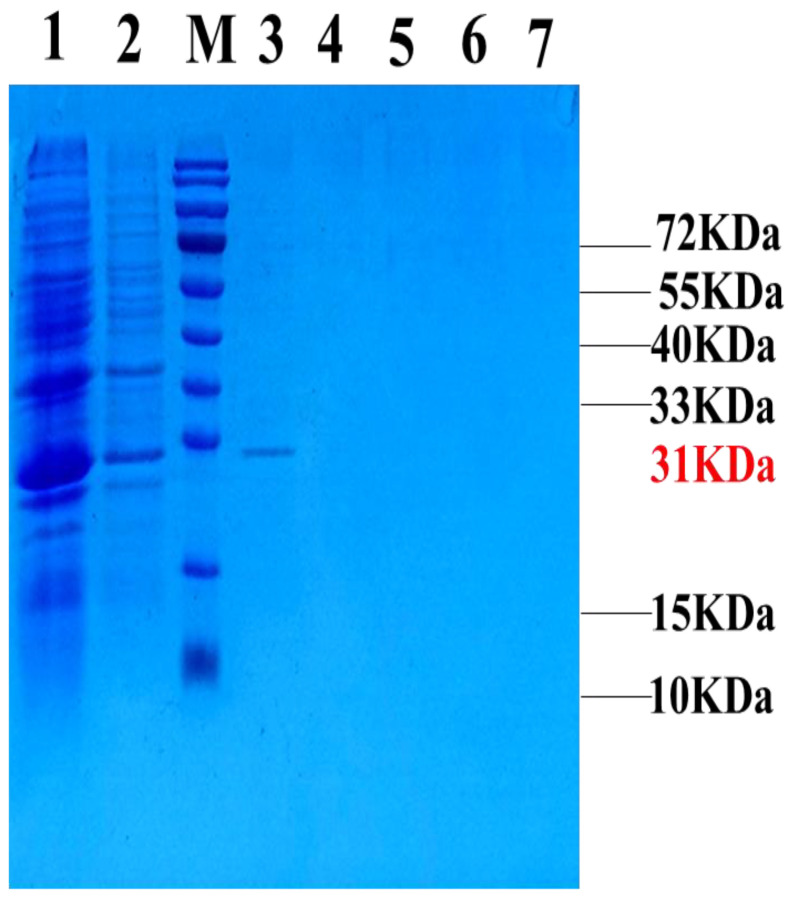
SDS-PAGE gel electrophoresis result of fusion protein GFP-LRGG purification. Lane 1: Before purification; lane 2: Flow through; M: Molecular weight standard; lanes 3–7: elution of 300 mM Imidazole.

**Figure 3 ijms-25-07072-f003:**
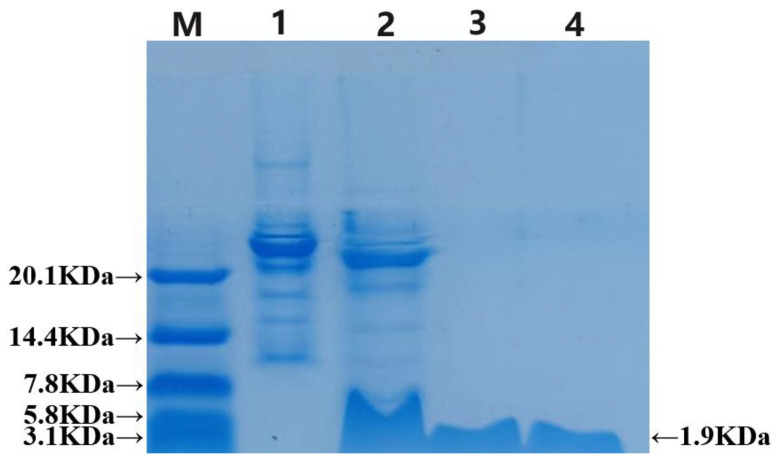
Tricine-SDS-PAGE gel electrophoresis result of LR_GG_, with cleavage by TEV enzyme. M: Protein marker (3.3–20.1 kD, Solarbio life sciences, Beijing, China). 1: Before cutting; 2: After TEV enzyme cutting; 3: Purified LR_GG_ of TEV enzyme cutting; 4: Chemically synthesized of LR_GG_.

**Figure 4 ijms-25-07072-f004:**
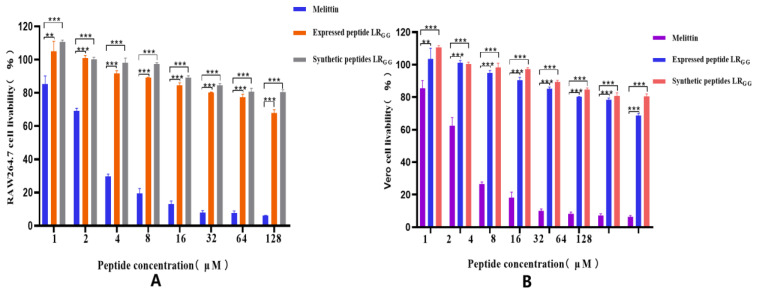
Cytotoxicity of the expressed peptide LR_GG_ and the synthetic peptide LR_GG_ to RAW264.7 (**A**) and Vero (**B**) cells. Note: *** *p* < 0.001 indicates a very significant difference, ** *p* < 0.01 indicates a very significant difference.

**Figure 5 ijms-25-07072-f005:**
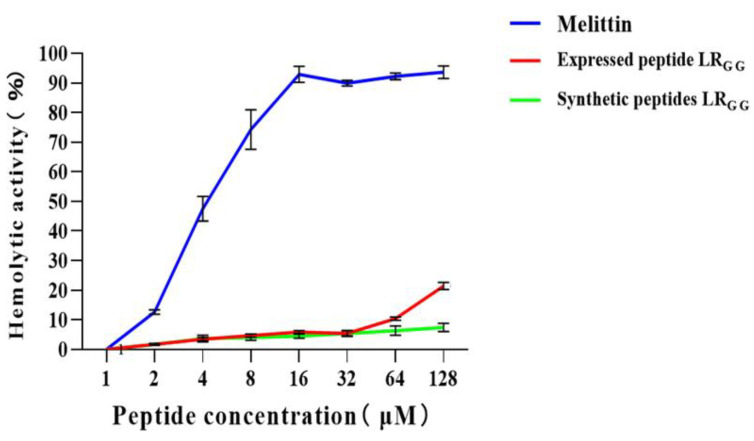
Hemolytic activity of the expressed peptide LR_GG_ and the synthetic peptide LR_GG_ on sheep red blood cells.

**Figure 6 ijms-25-07072-f006:**
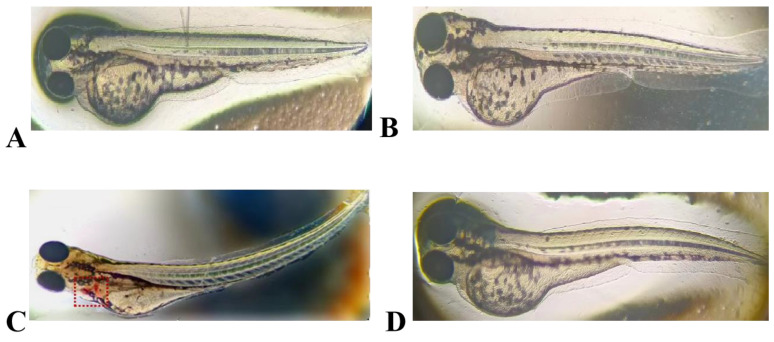
Embryonic safety of the expressed peptide LR_GG_ and the synthetic peptide LR_GG_ in zebrafish. (**A**) Negative control containing culture medium; (**B**) Containing 1 × Expression peptide LR_GG_ at MIC concentration; (**C**) Containing 200 μg/mL sodium dehydroacetate; (**D**) Containing 1 × MIC of the Chemical Synthesis Peptide LR_GG_.

**Figure 7 ijms-25-07072-f007:**
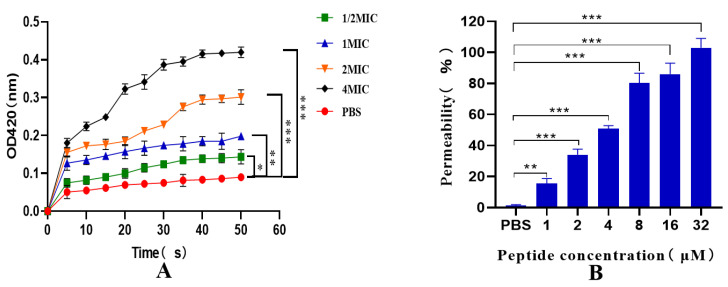
Pairs E. expressing peptide LR_GG_ Inner (**A**) and Outer (**B**) Membrane Permeability of *Eschericha coli* ATCC25922. Note: *** *p* < 0.01, there is a very significant difference, ** *p* < 0.01, there is a very significant difference, * *p* < 0.05, there is a significant difference.

**Figure 8 ijms-25-07072-f008:**
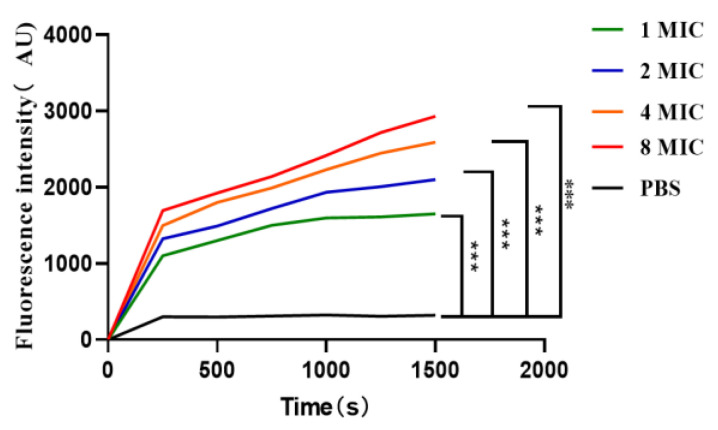
Effect of the antimicrobial peptide LR_GG_ on the plasma membrane potential of *Eschericha coli* ATCC25922. Note: *** *p* < 0.01, there is a very significant difference.

**Figure 9 ijms-25-07072-f009:**
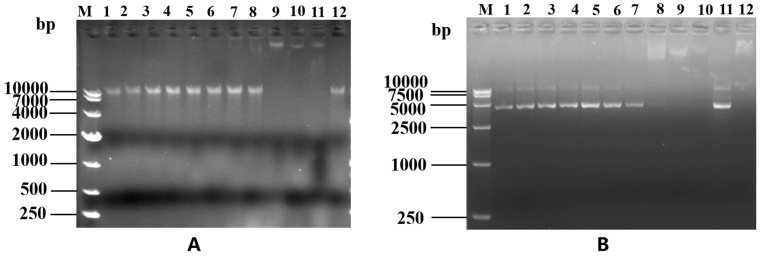
EMSA of genomic DNA (**A**) and plasmid DNA (**B**). (**A**) The effect of LR_GG_ on the genomic DNA of *Eschericha coli* ATCC 25922 EMSA test. M: DL10000 DNA Marker; 1–10: Gradually increasing concentrations of LR_GG_ peptide (1–512 μM); 11: Positive control; 12: Negtive control. In addition, all 1–12 with *Eschericha coli* ATCC 25922 genome (300 ng/μL). (**B**) Effects of different concentrations of LR_GG_ on the pkk3535 plasmid EMSA test. M: DL10000 DNA Marker; 1–10: Gradually increasing concentrations of LR_GG_ peptide (1–512 μM); 11: positive control; 12: Negtive control. In addition, all 1–12 with pKK3535 plasmid (300 ng/μL).

**Figure 10 ijms-25-07072-f010:**
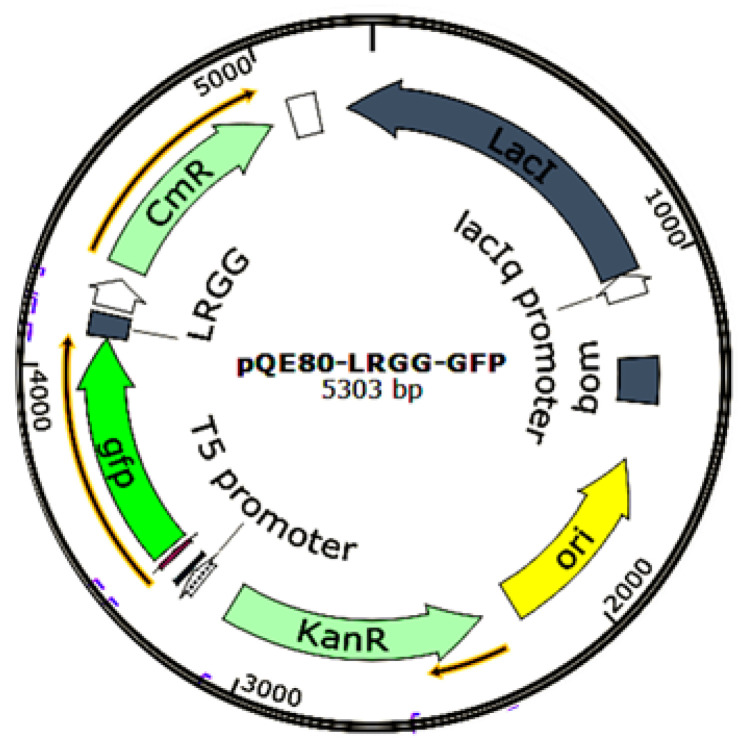
PQE-GFP-LR_GG_ plasmid map.

**Table 1 ijms-25-07072-t001:** MICs of the antimicrobial peptide LR_GG_ against bacteria (μM).

Test Strains	MICs (μg/mL)		
	Chem. Syn. LR_GG_	Expressed LR_GG_	Fusion Protein GFP-LR_GG_
Gram-negative			
*Eschericha coli* ATCC25922	2	2	›512
*S. pullorum* NCTC5776	4	4	›512
*K. pneumoniae* ATCC46117	8	16	›512
*P.aeruginosa* ATCC27853	8	8	›512
*S.flexneri* CMCC51572	8	8	›512
Gram-positive bacteria			
*Staphylococcus aureus* ATCC25923	32	32	›512
*S. aureus* ATCC29213	16	16	›512
*Enterococcus faecalis* ATCC29212	32	32	›512
MRSA	256	128	›512

**Table 2 ijms-25-07072-t002:** MIC value of the antimicrobial peptide LR_GG_ against *Eschericha coli* ATCC25922 at different temperatures and pH values.

AMPs	Control (pH 7)	Temperature			pH			
		0 °C	37 °C	100 °C	pH 4	pH 6	pH 8	pH 10
Chem. syn. LR_GG_	2	2	2	8	2	2	4	4
Expressed LR_GG_	2	2	4	8	2	4	2	16
Melittin	1	1	1	2	2	1	1	2

**Table 3 ijms-25-07072-t003:** MIC values of the antibacterial peptide LR_GG_ against *Eschericha coli* ATCC25922 under the action of different enzymes.

Peptide	Control	Proteinase (1 mg/mL)			
		Trypsin	Pepsin	Papain	Protease K
Chem. syn. LR_GG_	2	>128	>128	>64	>64
Exprssed LR_GG_	4	>128	>128	>64	>64
Melittin	2	4	4	2	4

**Table 4 ijms-25-07072-t004:** MIC values of the antimicrobial peptide LR_GG_ against *Eschericha coli* ATCC25922 under different salt ion environments.

Peptide	Control	Physical Salt Concentration				
		CaCl_2_	NaCl	KCl	NH_4_Cl	MgCl_2_
Chem. Syn. LR_GG_	2	4	8	2	2	4
Expressed LR_GG_	4	4	8	8	4	8
Melittin	2	4	4	2	2	2

Concentration of Salt Ion Solution: 2.5 mM CaCl_2_, 150 mM NaCl, 4.5 mM KCl, 6 mM NH_4_Cl, 1 mM MgCl_2_, 8 mM ZnCl_2_.

**Table 5 ijms-25-07072-t005:** MIC values of the antimicrobial peptide LR_GG_ against *Eschericha coli* ATCC25922 at different serum concentrations.

Peptide	Control	Serum				
		5%	10%	20%	40%	50%
Chem. syn. LR_GG_	2	2	2	8	16	16
Expressed LR_GG_	2	2	4	32	32	32
Melittin	2	32	128	128	128	128

**Table 6 ijms-25-07072-t006:** Primers used in this study.

Gene	Primer	Sequence (5′-3′)
linear pQE Vector	pQE-VT-F	GTAAAAGCTTAATTAGCTGAGCTTGGACTCC
	pQE-VT-R	CATATCTCTAGAGGATCCGTGATGGTG
GFP	GFP-F	CTAGAGATATGCGTAAAGGAGAAGAACTTTTCACTG
	GFP-R	AAGATTCTCATACTTGTATAGTTCATCCATGCCATGTGTAATCCC
LR_GG_	LR_GG_-F1	CTTACAGCAGACGCAGCAGACGACGGCCGCCACGACGCAG
	LR_GG_-R1	GAGAATCTTTATTTTCAGGGCCTGCTGCGTCTGCTGCGTCGTGGCGGC
TEV cleavage site + LR_GG_	TEV-LR_GG_-F2	GAGAATCTTTATTTTCAGGGCCTGCTGCGTCTGCTGCGTCGTGGCGGC
	TEV-LR_GG_-R2	TTACAGCAGACGCAGCAGACGACGGCCGCCACGACGCAG
validation primers	M13-F	AGGGTTTTCCCAGTCACG
	M13-R	GAGCGGATAACAATTTCACAC
	pQE30+	GTGAGCGGATAACAATTTCAC
	pQE30−	CTGAACAAATCCAGATGGAG

**Table 7 ijms-25-07072-t007:** Strains and plasmids.

Strains	Source
*Eschericha coli* stellar component cell	Takara
Transetta component cell	Takara
*Eschericha coli* ATCC 25922	Preserved by the Pharmacology and Toxicology Laboratory of Jilin Agricultural University
*S. pullorum* NCTC5776	
*K.Pneumoniae* CMCC 46117	
*P.aeruginosa* ATCC27853	
*S.flexneri* CMCC51572	
*S.aureus* ATCC 25923	
*S. faecalis* ATCC 29212	
Plasmids	
pQE-80-Kan	Qiagen
pMD-18T	Takara
pTZ18U-GFP	Takara

## Data Availability

The datasets supporting the conclusions of this article will be made available by the authors without undue reservation.

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
