# Peer review of "Prokaryotic Expression and Functional Verification of Antimicrobial Peptide LRGG"

_ijms, 2024, doi:10.3390/ijms25137072_

Round 1

Reviewer 1 Report

Comments and Suggestions for Authors

This paper presents a comprehensive report on the production, manufacturing, and efficacy of a newly developed antimicrobial peptide, LRgg. The significance lies in the fact that it is an invention directly from the lab level, rather than through an external agency, which adds considerable value. The data presented also hold substantial potential for breakthroughs. The methodology and storyline are robust, and the references are relatively appropriate. However, to enhance the paper, I have several essential suggestions regarding the methods, background, and data interpretation. Implementing these suggestions will enhance the clarity and depth of the paper, making it more comprehensive and impactful.

1. Ensure to provide the etymology of the "GG" subscript in the peptide name.

2. Page 1, line 22: Clarify why the figure 99.99% is used and the reasoning behind excluding 0.01%.

3. Add a section in the introduction part that discusses the benefits and drawbacks of antimicrobial peptides due to their peptide nature.

4. Explain why stellar component cells were advantageous for this experiment in the main text.

5. Detail why the tagging used with the specific vector was more beneficial compared to other tagging methods.

6. It is recommended to crop the lower part of Figure 4 to eliminate any smudged areas.

7. Figure 6: Justify the claim that a 20% hemolytic activity is not toxic and provide supporting evidence.

8. Given its importance, Figure 7 should be enlarged for better visibility and impact.

9. The Conclusion and Discussion sections should be reordered, placing the Discussion before the Conclusion.

Reviewer 2 Report

Comments and Suggestions for Authors

The article by Liu et al. deals with the expression in E. coli of an antimicrobial peptide already known to be active in Gram-negative bacteria.  In principle, the interest of this paper is to overcome chemical synthesis or purification, since both are high-cost procedures. The research seems well developed and includes fusion with a fluorescent protein that avoids the toxic effect of the peptide and that can ultimately be eliminated by enzymatic digestion.

Abreviations should be defined when used fisrt time e.g. NPN.

Legend of figures 3 and 4 should be rewritten (e.g. 4: chemically synthesized LRGG ?)

Some sentences are misleading (e.g."The peptide's primary mode of action involves disrupting the Gram-negative bacteria's cell membrane, leading to their death" which one the outer or plasma membrane? 

I believe the conclusion should appear after the discussion. 

One of the main concerns is related to the yield of the molecule by the engineered bacterium. This would greatly increase the quality of the article aslthough may also be a new biotechnological paper .

Round 2

Reviewer 2 Report

Comments and Suggestions for Authors

I think the ms. may be published.